# Psychological Tasks Used in Neuroimaging with Male Perpetrators of Intimate Partner Violence against Women: A Systematic Review

**DOI:** 10.3390/ijerph19158971

**Published:** 2022-07-23

**Authors:** Natalia Bueso-Izquierdo, Judit Caro Cabeza, Carlos Barbosa-Torres, Mónica Guerrero-Molina, Juan Manuel Moreno-Manso

**Affiliations:** 1Department of Psychology, University of Extremadura, 06006 Badajoz, Spain; nbueso@unex.es (N.B.-I.); monicagm@unex.es (M.G.-M.); jmmanso@unex.es (J.M.M.-M.); 2University of Extremadura, 06006 Badajoz, Spain; jucaroc@alumnos.unex.es

**Keywords:** intimate partner violence, neuroimaging, psychological tests, abusers

## Abstract

This bibliographic review analyses the utility for psychologists of using neuroimaging tests and psychological or neuropsychological tests at the same time for studying the functioning of the brain in male abusers condemned for intimate partner violence against women (IPVAW). So as to be able to find an answer, we reviewed the available studies that investigated the structure or functioning of the brain. The results of these reviewed works of research show the benefits of using neuroimaging applied to male abusers, together with the use, either simultaneously or not, of other types of psychological, neuropsychological, or observational tests to complement and/or amplify the results of the neuroimaging techniques, as this can help us to advance in the knowledge of neuroscience as concerns the mind of the male abuser.

## 1. Introduction

IPVAW is a serious public health problem that includes widely differing forms of violence involving high death rates [1,2]. The different forms that can be observed in this type of violence include physical, sexual, and psychological violence, as well as harassment [3]. This type of violence is more prevalent in the lives of women than some problems of a different nature, such as diabetes or depression, even though it is often not recognised or detected by health professionals [1]. Therefore, this review makes reference to the importance of the health system in early detection and the development of different policies for prevention and addressing them.

Taking into account the fact that it is a serious problem on a worldwide level, it is possible to find numerous studies that have focused on investigating the aggressor, i.e., the male abusers, and how and why they commit acts of violence against their partners [3,4,5,6]. Within this type of violence, we can find different typologies of male abusers; among the most commonly cited in the literature are three types of male abuser. The first is the *family-only* abuser, whose abuse and aggressiveness is perpetrated solely toward members of his own family; it is highly unlikely that these abusers will be aggressive or commit some illegal act outside the home. The acts that they commit which can be categorised as intimate partner violence would probably include both sexual and psychological abuse. Then, we have the *dysphoric or borderline* abusers, who commit acts of violence against a woman that are more severe than those of the first group; in addition, although their abuse is generally within the family, they can commit extra-familiar violence and may include some kind of criminal behaviour. These men are emotionally volatile, with schizoid or breaking-point personality traits, and they probably have problems of alcohol or drug abuse. Last of the three, we have generally *violent or antisocial* abusers, who would perpetrate severe abuse of their partner but would also be involved in extra-familiar violence in a much wider sense than the other two groups. They could also frequently behave in a criminal way and may fall foul of the law. In addition, they may have alcohol- and/or drug-related problems, as well as presenting antisocial or psychopathic personality traits [7].

Other categories are currently under study, with the proposal of only two types of male abuser: the specialists and the generalists [8]. In this sense, the *specialist* abusers would be those whose crimes are related to violence aimed solely at their intimate partner; while the *generalist* abusers would be delinquents with criminal records for various types of crime, including violence against their intimate partners.

The studies reveal that the abuse perpetrated by these males may be due to various causes, which may not be the direct cause but could facilitate the said violence. Such factors could include experiencing violence as children [9,10,11], a low level of satisfaction in the relationship [12], alcohol and drug abuse [13,14,15], suffering from attention deficit hyperactivity disorder (ADHD) [16], problems in their executive functions [17], or brain damage [18], among many others. 

Furthermore, these deficits in the executive functions only seem to appear in this type of population as compared to other delinquents or other nondelinquent males [3,17]. Nevertheless, the executive functions that have been studied with respect to male abusers are not the only variables at a neuroscientific level that have been analysed in this population. In fact, there are several works of research that have checked the functional and structural differences in the brains of male abusers who commit intimate partner violence and other males who show criminal behaviour patterns or not [19,20,21,22].

The psychological and neuropsychological tests allow us to discover which characteristics these men have. They also allow us to see what their motivations are, the objectives they pursue through their actions, and their perceptions, emotions, and beliefs, as well as their level of intelligence or the working of their emotional processes and their executive functions [23]. On the other hand, the neuroimaging tests detect the regions of the brain that help to better understand these beliefs, how these regions, which are affected in different ways, also affect the reasoning concerning their objectives and their actions, and how they are involved in inhibitory control or in syntactic processing [23].

Thus, combining both types of tests in the same study could help us to discover whether they are closely related. Since the tests based on questionnaires or instruments show us the variables concerning behaviour and the mechanisms of perceptual, cognitive, and motor control, as well as explaining the necessary components for carrying out a task, neuroimaging can reveal which specific areas of the brain are activated by certain cognitive or behavioural tasks [24,25,26]. As both types of test correlate and complement each other, they can be a useful joint tool for professionals in psychology. This would allow them not only to know the psychological processes of the subject under study, but also to provide an explanation of the said functioning.

The current bibliographic review analyses the existing studies whose objective is to study the functioning or structure of the brains of male abusers condemned for IPVAW using neuroimaging tests which are complemented by psychological and/or neuropsychological tests to analyse the differences and/or similarities in the behaviour of members of this population and of other groups made up of criminals or nondelinquent males.

## 2. Method

### 2.1. Study Design

This systematic review was performed under the recommended reference framework of Preferred Reporting Items for Systematic Reviews and Meta-Analyses (PRISMA) guidelines. The search was based on the following databases: Google Scholar, Pubmed, SCOPUS, and ScienceDirect, up to 2000–2022. 

### 2.2. Search Strategy

The algorithm used for all the databases was the following: *“neuroimaging”* and *“abusers”* and *“neuroimaging”* and *“perpetrators”* between the years 2000 and 2022, according to data from Google Scholar. We found a total of 180 results using the terms “neuroimaging” and “abusers”, and 7150 results using the terms “neuroimaging” and “perpetrators”, “batterers”, and “abusers”.

After having conducted a global and a preliminary search in Google Scholar, we examined the following databases: Pubmed, Scopus, and ScienceDirect. The terms used were “neuroimaging”, “perpetrators”, “batterers”, and “abusers” between the years 2000 and 2022. 

The search was carried out using the logic operator AND to relate the above descriptors; hence, the final search was “neuroimaging” AND “perpetrators”, “neuroimaging” AND “batterers”, and “neuroimaging” AND “abusers”. Concerning the term “neuroimaging and perpetrators”, we found 39 results in Pubmed, while ScienceDirect gave us 595; in Google Scholar, we found 7150 articles, while Scopus revealed 31. As for the search “neuroimaging and batterers”, we found 25 results in Pubmed, 146 in ScienceDirect, 486 in Google Scholar, and five in Scopus.

The results obtained using “neuroimaging and abusers” were 4241 in Pubmed, 20,892 in ScienceDirect, 20,000 in Google Scholar, and 112 in Scopus. Finally, applying the term “neuroimaging and batterers”, no results were obtained in Pubmed, in contrast to one in ScienceDirect, 180 in Google Scholar, and again none in Scopus. The entire selection process is shown in Figure 1.

### 2.3. Selection Criteria

The search included all articles published since 2000 to allow for an adequate review of the literature. All the articles were published in English and, until the time of drafting this article (March 2022), were considered eligible for this review. The other inclusion criteria were studies concerning (a) males condemned for crimes involving IPVAW or males who had been reported for abuse of their female partners, (b) the use of neuroimaging techniques; the articles also had to be quantitative. 

Using a snowballing technique, references of eligible studies and relevant reviews were also searched. Two authors (N.B. and J.C.), working independently of each other in pair, performed the selection of studies.

### 2.4. Data Extraction and Analysis

A piloted data extraction form was used to extract data from eligible articles, which were reviewed simultaneously and independently by two reviewers (C.B.T. and M.G.M.). The following data were extracted from each study: title of the article, name of authors and year of publication, study design, selection of sample, outcomes, statistical analysis, and main conclusions. Discussion by reviewers and team consensus resolved any disagreement.

## 3. Results

### 3.1. Description of Selected Studies

Following these final criteria, there were approximately 50–60 results. Following an exhaustive examination, a final total of 14 articles included and reviewed was obtained (Table 1 and Table 2). 

### 3.2. Studies That Simultaneously Applied Neuroimaging and Another Test

First of all, some studies described the use of neuroimaging techniques, as well as other tests of a psychological, neuropsychological, or observational type. We found several research articles that used observational techniques while also applying neuroimaging; thus, these results can be divided into studies that applied observational techniques together with neuroimaging and studies that applied other types of tests.

#### 3.2.1. Studies with Observational Tests

In the first study that used neuroimaging to analyse the brain of male abusers while also applying an observational technique for images, they subjected a sample of 10 male abusers and 13 other males, as a control group, to four types of visual stimuli: neutral, positive affection, aggressive threat, and aggression aimed at a woman. The results show that male abusers presented a significantly greater hyper-neuronal response to threat stimuli in the hippocampus, the fusiform gyrus, the posterior cingulated cortex, the thalamus, and the occipital cortex; this would be interpreted as an irregularity in affective processing [20].

Following the lines of the above study, we found another research article that also used images while applying neuroimaging [24]. In this case, the sample was divided into males condemned for intimate partner violence and males condemned for other crimes; the images were divided into three categories: images of intimate partner violence, images of general violence, and neutral images. The results indicated the activation of the posterior cingulated cortex as a response to images related to aggression against women, as well as the activation of the insular cortex and the precuneus on observing images related to violence (threatening stimuli), taking into account the fact that these are key structures regarding aggressive behaviour. In addition, the results replicated to a great extent those obtained in the abovementioned study of Lee et al. [20].

Another research work that applied an observational technique, but with a different sample in comparison to the others, is that of Flanagan et al. [25], in which a sample made up not only of males, but also of females, used a series of videos during the application of the neuroimaging. The experiment consisted of playing six videos of 3 min in duration each: three of one category and the other three of the other, played alternately. The first category was videos of arguments between couples, while the other was videos that described a normal morning routine. With regard to IPVAW, the results of this study indicated that, for male abusers, there was a greater activity in the right amygdala during the neutral signals, while an irregularity was once again observed in the emotional component with respect to the aggressive behaviour.

Marín-Morales et al. [27] also used an observational technique while neuroimaging. It was also the first study to examine the neural bases regulating emotions in male abusers. In this study, as in those of Lee et al. [20] and Bueso-Izquierdo et al. [24], an observational exercise concerning images was also carried out while applying the neuroimaging technique. The images were divided into three categories: neutral images, images with a negative valence, and images related to intimate partner violence, while the last category had images used by Bueso-Izquierdo et al. [24]. However, in this study, in addition to showing the images, they were also given three types of instructions depending on the photograph: “observe” (look at the image passively), “experience” (put yourself in the place of the person in the photograph), and “suppress” (control the negative emotions that the image may arouse through regulating strategies previously taught). The results correspond to and are in line with those obtained by the two previously cited studies.

To summarise, in all these works of research, it is possible to analyse how the observational tests complemented the neuroimaging, revealing how certain areas of the brain were activated when shown images relating to IPVAW; this allowed us to verify the areas that have the closest relation to IPVAW. 

#### 3.2.2. Studies with Other Types of Tests

In this category, we can find that the observation techniques are not the only ones being used simultaneously to study the functioning of the brain in male abusers. First of all, the research carried out by Stanford et al. [17] divided a sample into male abusers and a control group. The subjects were given the auditory oddball task, a test in which the subjects are given a series of identical audio stimuli and, now and again, one that is different, which allows the amplitude of the P3 to be observed. The results of the auditory oddball task showed a lower amplitude of P3 related, according to the authors, to previous studies using participants with an antisocial personality disorder and impulsive–aggressive individuals, since these individuals also showed a low amplitude of P3, thus correlating with problems of behaviour disinhibition.

On the other hand, Lee, Chan, and Raine [28], with a sample divided in the same way as Stanford et al. [17], used neuroimaging tests while also applying emotional and cognitive tests, such as the Stroop test. The results in the Stroop test indicated that male abusers respond more slowly to negative affection stimuli than neutral persons. As for the results of the neuroimaging, less frontal activation could be seen when faced with aggressive stimuli, whereas a greater limbic activation could be seen, thus coinciding with the results of [20,21,24,25].

Similar results were found in the study of Marín-Morales et al. [29], who did not use observation techniques either, rather giving the subjects a series of dilemmas during the neuroimaging. The test consisted of a total of 40 dilemmas, belonging to five different categories, with three of them belonging to the task of Greene et al. [30]: personal, impersonal, and control dilemmas. The other two categories were conditions designed specifically for this study: moral dilemmas of general violence and specific dilemmas of intimate partner violence. In each dilemma, it was necessary to decide to be violent or not toward a person; in those of general violence, it involved a person they knew (a family member, a neighbour, etc.), while, in those of intimate partner violence, the partner was involved. 

The results showed that male abusers did not activate the areas of the brain connected with moral decision making when the dilemmas concerned intimate partner violence; however, they did do so in the case of general violence, unlike the group made up of other criminal types, when the said areas of the brain were activated in both cases. The activated regions were the prefrontal cortex, the precuneus, the posterior cingulated cortex, and the angular gyrus, coinciding with the results obtained in the previously analysed studies.

What is noticeable from all these studies (Table 1) that applied a test simultaneously with neuroimaging is that they easily allow comparisons to be made of the areas that are activated in specific conditions. It is interesting to be able to compare whether or not a particular area is activated in the same way when faced with, for instance, general violence and intimate partner violence, as well as the differences across male abusers, other delinquents, or subjects who are not delinquents under the various conditions. It should be pointed out that most of the works of research coincided in their results with few differences concerning the activation or not of certain areas of the brain. In short, above all, we found that such areas as the cingulated cortex, the insular cortex, the precuneus, and the amygdala were activated, with a certain relation to emotional activation when faced with stimuli concerning intimate partner violence.

**Table 1 ijerph-19-08971-t001:** Articles, authors, psychological tests, and results obtained in studies with neuroimaging and another test used simultaneously.

Title of the Article	Bibliographic Reference	Setting	Type of Test Used	Result Obtained
**P3 amplitude reduction and executive function deficits in men convicted of spousal/partner abuse**	Stanford et al. (2007). P3 amplitude reduction and executive function deficits in men convicted of spousal/partner abuse. *Personality and Individual Differences*, 43(2), 365–375. [17]	Cross-sectional not randomised study with control group	Trail Making Test, Wisconsin Card Sorting Test, EEG, and auditory oddball task	Male abusers showed a lower amplitude in P3, as well as important cognitive deficits in impulse control and executive functions.
**Strong limbic and weak frontal activation to aggressive stimuli in spouse abusers**	Lee et al. (2008). Strong limbic and weak frontal activation to aggressive stimuli in spouse abusers. *Molecular Psychiatry*, 13(7), 655–656. [28]	Cross-sectional not randomised study with control group	fMRI and emotional and cognitive tests from Stroop	Male abusers showed a greater limbic activation and less frontal activation to aggressive stimuli.
**Hyperresponsivity to threat stimuli in domestic violence offenders: a functional magnetic resonance imaging study**	Lee et al. (2009). Hyperresponsivity to threat stimuli in domestic violence offenders. *The Journal of Clinical Psychiatry*, 70(1), 36–45. [20]	Cross-sectional not randomised study with control group	fMRI and an observation test with images	Male abusers showed a significantly greater neuronal hyperresponsivity to threat stimuli in the hippocampus, the fusiform gyrus, the posterior cingulated cortex, the thalamus, and the occipital cortex.
**Are batterers different from other criminals? An fMRI study**	Bueso-Izquierdo et al. (2016). Are batterers different from other criminals? An fMRI study. *Social Cognitive and Affective Neuroscience*, *11*(5), 852–862. [24]	Quasi-experimental cross-sectional study with comparison of groups	fMRI and *n* observation test with images	Male abusers showed the activation of the orbitofrontal cortex and the posterior cingulated cortex, as well as the deactivation of the anterior cingulated cortex and the insular cortex, when observing images related to intimate partner violence.
**Preliminary development of a neuroimaging paradigm to examine neural correlates of relationship conflict**	Flanagan et al. (2019). Preliminary development of a neuroimaging paradigm to examine neural correlates of relationship conflict. *Psychiatry Research: Neuroimaging*, 283, 125–134. [25]	Cross-sectional not randomised study with comparison of groups	fMRI and the viewing of 6 videos	A positive correlation was observed between a neutral interaction among male abusers and their partners/spouses and the activation of the right-hand part of the amygdala.
**“Would you allow your wife to dress in a miniskirt to the party?”: Batterers do not activate default mode network during moral decisions about intimate partner violence**	Marín-Morales et al. (2020). “Would you allow your wife to dress in a miniskirt to the party?” *Journal of Interpersonal Violence*, 37(3–4). [29]	Cross-sectional study with sample by suitability with comparison of groups	fMRI while resolving a series of dilemmas	When male abusers are presented with dilemmas related to intimate partner violence, the regions of the brain related to moral decision taking are not normally activated, but they are activated when facing dilemmas related to general violence.
**Emotional regulation in male batterers when faced with pictures of intimate partner violence. Do they have a problem with suppressing or experiencing emotions?**	Marín-Morales et al. (2021). Emotional regulation in male batterers when faced with pictures of intimate partner violence. Do they have a problem with suppressing or experiencing emotions? *Journal of Interpersonal Violence*. [21]	Cross-sectional not randomised trial with control groups	fMRI and an observation test with images	Male abusers show a greater activation in the parietal cortex, the prefrontal cortex, the anterior cingulated cortex, and the insular cortex when observing images related to intimate partner violence.

### 3.3. Studies Applying Neuroimaging Independently of Other Tests

In this section, we look at studies investigating the brain of male abusers with neuroimaging techniques, but without the assistance of other psychological or neuropsychological tests during the application of this technique. Some of these works of research used other tests besides neuroimaging, albeit not at the same time. They used them as a tool to complement the neuroimaging results.

First of all, the study of George et al. [31] divided the participants into abusers who also suffered from alcoholism, non-abusers who were also alcoholics, and non-abusers who were not alcoholics either. The objective was to measure the glucose metabolism in the brain using positron emission tomography with fluorodeoxyglucose (FDG PET). The results showed that the reuptake of glucose by abusers in the hypothalamus was lower and that there were fewer correlations between numerous cortical and sub-cortical structures in comparison with the other two groups; it is worth noting the scarcity of connections between the amygdala and various cortical structures, which are very important for control and fear-induced aggression. This, in fact, could lead us to understand that, on a psychological level, these men had problems keeping themselves under control in situations where they felt threatened. With the sample divided into the same groups as in the study of George et al. [31], Zhang et al. [32], who subjected the participants to MRI, also obtained similar results. In their results, the male abusers, in comparison with the other two groups, showed a very reduced volume in the right amygdala.

Expanding on the results obtained by Zhang et al. [32], we find the study by Verdejo-Román et al. [22], in which a lower volume was observed, in addition to in the right amygdala, in the posterior and anterior cingulated cortex, the parahippocampal gyrus, the insular cortex, and the orbitofrontal cortex. In addition to the MRI, the test of the 60 faces of Ekman was also later applied. This test concerns emotional perception and allows a meaning to be given to the results obtained from the neuroimaging. In these results, a lower volume was observed in the posterior cingulated cortex related to worse scores in the Ekman test.

Another recent study was that of Bueso-Izquierdo et al. [19], in which only the MRI was applied, in order to observe whether there was any type of abnormality in the cerebral structure of these three groups of participants: one made up of male abusers, another of males who had committed other crimes, and a third control group. Although 14 of the participants showed abnormalities in the cerebral structure, it was considered that the majority were not clinically relevant and that they were not specifically related to intimate partner violence; therefore, they were not responsible for the differences in the cerebral functioning among abusers and other delinquents found in previous studies.

Concerning the volume of certain cerebral structures and the emotions, we also found the studies of Marín-Morales et al. [27] and Amaoui et al. [33]. Both these studies had samples divided into males condemned for intimate partner violence, males not condemned for any crime, and males condemned for crimes other than intimate partner violence. The subjects of both studies were also subjected to MRIs.

In the first study of Marín-Morales et al. [27], apart from the MRI, the participants had to answer the Cognitive Emotion Regulation Questionnaire (CERQ), while, in the study of Amaoui et al. [33], in addition to the CERQ, the Emotion Regulation Questionnaire (ERQ), the Difficulties in Emotion Regulation Scale (DERS), and the Interpersonal Reactivity Index (IRI) were applied. The results showed that the cerebral volume of certain structures was related to maladapted emotion regulation strategies in male abusers. However, the results of the study of Marín-Morales et al. [27] showed no difference between male abusers and other delinquents; therefore, they suggest that the lower volume in the cerebral regions related to emotion regulation are in fact related to their criminal behaviour.

Lastly, we also found the study of Amaoui et al. [33], which had a sample divided into the same groupings as in the studies of Marín-Morales et al. [34] and Marín-Morales et al. [21]. In this study, the fMRI in a state of repose (rs-fMRI) was used to study the involvement of the Crus II zone of the cerebellum in mentalisation (this is the ability people have to understand and to think about the mental states of others). In addition to neuroimaging, they used some other tests apart from the MRI so as to be able to complement the results of the neuroimaging. These instruments were the Interpersonal Reactivity Index (IRI), the Inventory of Distorted Thoughts on Women and Violence (IDTWV), and the Eyes Test. In the results, we found functional connectivity between the Crus II and various areas of the brain, with the precuneus, which correlates positively with having irrational thoughts about women, with the half-temporal gyrus, which correlates negatively with empathy, and with the DMN (default mode network), which is related to difficulties male abusers have for processing emotions. A greater connectivity was related to a higher number of distorted thoughts about women and a lower empathy. Last but not least, there was also connectivity between the para-hippocampus and the hippocampus, which could be related to difficulties with the moral processing of emotions. Thus, we once again found cerebral irregularities that correspond to difficulties with processing the emotions.

In short, what we can see in the results of these studies (Table 2) are the very characteristics of the brain structure of male abusers and how these differences with the general population or with other types of criminal populations explain the behaviour of these men, particularly with respect to emotions and their regulation. Furthermore, the works of research that not only use neuroimaging, but complemented it with other tests, either at the same time or later, allow us to give some meaning to the neuroimaging, relating the results to other complementary tests. Accordingly, it is possible to understand that the lower volume in these cerebral structures or the scarcity of connections between them affects the way in which the subjects live their emotions and, thus, their behaviour.

**Table 2 ijerph-19-08971-t002:** Articles, authors, complementary tests (or not), and results obtained from the neuroimaging in studies using neuroimaging independently.

Title of Article	Bibliographic Reference	Setting	Type of Test Used	Result Obtained
**A select group of perpetrators of domestic violence: evidence of decreased metabolism in the right hypothalamus and reduced relationships between cortical/subcortical brain structures in position emission tomography**	George et al. (2004). A select group of perpetrators of domestic violence: evidence of decreased metabolism in the right hypothalamus and reduced relationships between cortical/subcortical brain structures in position emission tomography. *Psychiatry Research: Neuroimaging*, 130(1), 11–25. [31]	Cross-sectional not randomised trial with comparison groups	FDG PET	Alcohol-dependent abusers show a lower absorption of glucose in the hypothalamus and less correlation between the cortical and subcortical structures.
**Smaller right amygdala in Caucasian alcohol-dependent male patients with a history of intimate partner violence: a volumetric imaging study**	Zhang et al. (2011). Smaller right amygdala in Caucasian alcohol-dependent male patients with a history of intimate partner violence: a volumetric imaging study. *Addiction Biology*, 18(3), 537–547. [32]	Cross-sectional not randomised study with sample by suitability with comparison of groups	No complementary tests while neuroimaging carried out	The abusers have a lower volume in the right amygdala.
**Prevalence and nature of structural brain abnormalities in batterers: a magnetic resonance imaging study**	Bueso-Izquierdo et al., (2018). Prevalence and nature of structural brain abnormalities in batterers: a magnetic resonance imaging study. *International Journal of Forensic Mental Health*, 18(3), 220–227. [19]	Quasi-experimental cross-sectional study with comparison of groups	Nothing used during neuroimaging	Fourteen participants showed abnormalities in the brain structure. It was concluded that these abnormalities were not specifically related to intimate partner violence.
**“Structural brain differences in emotional processing and regulation areas between male batterers and other criminals: a preliminary study”**	Verdejo-Román et al., (2018). Structural brain differences in emotional processing and regulation areas between male batterers and other criminals: a preliminary study. *Social Neuroscience*, 14(4), 390–397. [22]	Cross-sectional study with sample by suitability with comparison of groups	MRI and Ekman’s 60 faces test	The abusers showed a finer cortex in the posterior and anterior cingulated cortex, the para-hippocampal gyrus, the insular cortex, and the orbitofrontal cortex. The lower thickness in the posterior cingulated cortex is related to worse scores in Ekman’s 60 faces test.
**P.0888 Strategies of emotional regulation and brain volume in male perpetrators and other criminals**	Marín-Morales et al., (2021). P.0888 Strategies of emotional regulation and brain volume in male perpetrators and other criminals. *European Neuropsychopharmacology*, 53, S651–S652. [27]	Cross-sectional not randomised trial with comparison group	MRI and CERQ	The regional cerebral volume is related to the use of maladapted emotion regulation strategies in male abusers.
**Social mentalising in male perpetrators of intimate partner violence against women is associated with resting-state functional connectivity of the Crus II**	Amaoui et al., (2022). Social mentalising in male perpetrators of intimate partner violence against women is associated with resting-state functional connectivity of the Crus II. *Journal of Psychiatric Research*, 150, 264–271. [33]	Cross-sectional not randomised trial with comparison group	rs-fRMI, IRI, IDTWV, and The Eyes Test	In abusers, the Crus II shows connectivity with the precuneus, the half-temporal gyrus, the DMN, the hippocampus, and the para-hippocampus. This affects the processing of emotions, empathy, and irrational thoughts about women.
**Lower brain volume and poorer emotional regulation in partner coercive men and other offenders**	Marín-Morales et al., (2022). Lower brain volume and poorer emotional regulation in partner coercive men and other offenders. *Psychology of Violence*, 12(2), 104–115. [34]	Cross-sectional not randomised study with comparison groups	MRI, ERQ, CERQ, DERS, and IRI	The abusers showed a lower volume in the right nucleus accumbens and in the left dorsal anterior cingulated cortex than the nondelinquent group. The lower volume in these structures is related to difficulties in regulating emotions and empathy.

## 4. Discussion

This bibliographic review provided a detailed account of the use of neuroimaging techniques and psychological or neuropsychological tests as tools to understand the way the brain of male perpetrators of intimate partner violence functions. It can be said that the information concerning the topic being dealt with is novel but insufficient, as there are still only a very few empirical studies looking into this subject. Thus, there is all the more reason to continue investigating it in the future. The idea would be to not only analyse the usefulness of other instruments, but also to try to replicate the studies already carried out so as to be able to verify the results with larger samples, as has been pointed out by other authors [35,36].

One of the main ideas that can be extracted is that, independently of the instruments or the techniques used in the different current works of research, the findings of each study reviewed correspond to the specific cerebral functions of male perpetrators of intimate partner violence with respect to the processing and regulation of their emotions. This problem would affect the control of emotions and impulses, aggressive behaviour, and a lack of empathy. In general, it is the same regions of the brain that are affected, particularly the cingulated gyrus, the precuneus, the insular cortex, or the amygdala, among others. In addition, we found that these areas, to a greater or lesser extent, are related to criminal behaviour, when compared with other males who are not perpetrators of IPVAW.

As for the instruments, independently of whether the reviewed studies used solely a neuroimaging technique or complemented it with another psychological or neuropsychological test, simultaneously or not, we found that the results concerning the structure of the brain were reasonably similar since, as already mentioned above, we found that the same zones were involved in almost all the works of research. Nevertheless, we also found that, as we were able to relate these results to other types of tests that acted as complements in these studies, we were able to find a psychological meaning for these differences in the various regions of the brain. For instance, in the case of tests or questionnaires that were not applied simultaneously, it was clear that they are useful for complementing the neuroimaging results, as they allow us to relate the structures found to the results of the questionnaires and to understand the meaning of a low score in a particular item with a greater volume in one area. 

On the other hand, it is worth noting that the studies which apply neuroimaging and another type of test simultaneously allow verifying the differences in the brain not only on a structural level, but also on a functional level. This is because, with the test they are submitted to in each study, we can verify the different levels of activation of the areas of the brain and check the said differences in activation against the circumstances of the male perpetrators, thus verifying which areas are related to intimate partner violence against women in this population. 

## 5. Conclusions

Both neuroimaging and the questionnaires or psychological and neuropsychological tests are extremely useful tools for this type of research, and they individually provide valuable knowledge. On the basis of the results from this review, we also consider these techniques to be very useful for psychologists when carrying out research, as they allow a detailed study of the objectives of the research, thus taking advantage of all the available resources and allowing us to achieve an explanation for the results obtained. With the results of a psychological test, we can know what is happening, whereas, with the results of a neuroimaging test, we can know why it is happening.

Similarly, it would be useful to look into how other types of psychological tests could be used to study the brain functions of male abusers, since, as pointed out above, there are currently few such studies, many of them using fairly similar tests, which means that we have few resources with which to work, and this supposes an inconvenience if we take into account the current size of the IPVAW problem.

On the other hand, other questions for future studies could involve working with larger samples, as this would allow us to observe a larger number of subjects and, thus, include different cultural, social, demographic, and genetic variables. This, in turn, would allow us to verify whether the results could be applied as a general rule for all male abusers condemned for intimate partner violence or whether it would be necessary to divide them into different typologies. Furthermore, we believe that it would be valuable, in future research, to carry out detailed interviews with the subjects along with a risk-of-violence questionnaire to evaluate the danger. It could also be beneficial to have subjects who, as much as possible, belong to a similar cognitive state when applying the different tests (controlling the intake of alcohol or other substances through prior analyses, performing beforehand some kind of relaxation technique, carrying out the study on similar timetables, etc.).

Lastly, this review was based on studies of the structure or functioning of the brain of male abusers, while still taking into account the fact that it is important to know that there are different facilitators that can predispose a person to exercising intimate partner violence. Furthermore, knowing how serious this problem is on a worldwide level, we should be alert in all spheres and continue to investigate all these questions, which can help us to detect, understand, and even find solutions and adequate treatments so as to prevent future cases of abuse/battering and stop them recurring.

## Figures and Tables

**Figure 1 ijerph-19-08971-f001:**
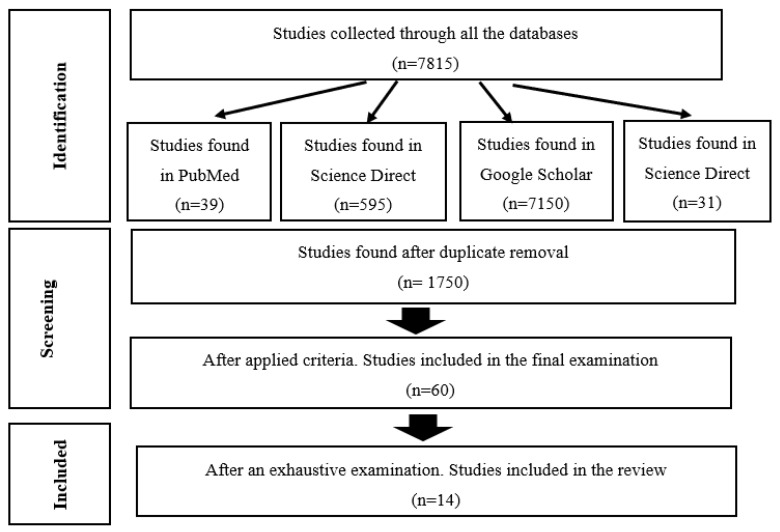
Flowchart of selection of studies.

## Data Availability

All data generated or analyzed during this study are included in this published article.

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
