# Peer review of "Psychological Tasks Used in Neuroimaging with Male Perpetrators of Intimate Partner Violence against Women: A Systematic Review"

_ijerph, 2022, doi:10.3390/ijerph19158971_

Round 1

Reviewer 1 Report

I would suggest to include the PRISMA flowchart, to clarify the different stages in selecting (or not) the studies during the process. I also strongly suggest to clarify the "search strategy" and the "selection criteria" sections. I could not understand how many articles you finally found with the "final" algorithm, and how many duplicates (if any) were obtained. 

Author Response

Thank you for your suggestion. We have added this flowchart to clarify our search strategy.

Reviewer 2 Report

Thank you for having me to review this work, and congratulations to the authors for its preparation, it is extremely interesting. In the attached document you will find my considerations on the matter, after an exhaustive reading and analysis. 

Best regards.

Author Response

Thank you for your suggestion. please see the attachment.

Round 2

Reviewer 2 Report

Accepted in present form.